# Influence of Ionomer and Cyanuric Acid on Antistatic, Mechanical, Thermal, and Rheological Properties of Extruded Carbon Nanotube (CNT)/Polyoxymethylene (POM) Nanocomposites

**DOI:** 10.3390/polym14091849

**Published:** 2022-04-30

**Authors:** Sang-Seok Yun, Dong-hyeok Shin, Keon-Soo Jang

**Affiliations:** 1Department of Polymer Engineering, School of Chemical and Materials Engineering, The University of Suwon, Hwaseong 18323, Korea; fourdai@suwon.ac.kr; 2Woosung Chemical Co., Cheonan-si 31214, Korea; sdh@metapoly.co.kr

**Keywords:** polyoxymethylene (POM), carbon nanotube (CNT), antistatic, ionomer, cyanuric acid, electrical properties, mechanical properties, thermal properties

## Abstract

The electrical properties of carbon-based filler-embedded polymer nanocomposites are essential for various applications such as antistatic and electromagnetic interference (EMI) applications. In this study, the impact of additives (i.e., ethylene-co-acid-co-sodium acid copolymer-based ionomer and cyanuric acid) on the antistatic, mechanical, thermal, and rheological properties of extruded multiwalled carbon nanotube (MWCNT)/polyoxymethylene (POM) nanocomposites were systematically investigated. The effects of each additive and the combination of additives were examined. Despite a slight reduction in mechanical properties, the incorporation of ionomer (coating on CNTs) and/or cyanuric acid (π-π interaction between CNTs and cyanuric acid) into the POM/CNT nanocomposites improved the CNT dispersity in the POM matrix, thereby enhancing electrical properties such as the electrical conductivity (and surface resistance) and electrical conductivity monodispersity. The optimum composition for the highest electrical properties was determined to be POM/1.5 wt% CNT/3.0 wt% ionomer/0.5 wt% cyanuric acid. The nanocomposites with tunable electrical properties are sought after, especially for antistatic and EMI applications such as electronic device-fixing jigs.

## 1. Introduction

Polyoxymethylene (POM; polyacetal; polyformaldehyde) is one of the most crucial engineering polymers [1]. It exhibits high crystallinity (>70%), owing to the flexible main chain comprising repeating –CH_2_–O– units, which results in an opaque white material and a high density (approximately 1.41 g/cm^3^). It features excellent short-term mechanical properties (tensile strength, toughness, and rigidity), a low tendency to creep and fatigue, and a low linear coefficient of thermal expansion [1,2,3]. It also has excellent stability (resistance to chemicals, organic solvents, and fuels at room temperature), low water permeability/absorption, good tribological properties (high hardness and glossy and smooth molded surfaces), and the tendency to maintain good electrical and mechanical properties at temperatures between −40 and −140 °C. They can be maintained at 90 °C for the long term and 140 °C for the short term [1,4,5,6,7,8]. Therefore, POM has been extensively utilized in a myriad of engineering applications such as bearings, gears, conveyer belt links, wear surfaces, creep resistant housings, gas caps, chemical sprayers, soap dispersers, paint mixing paddles, and safety systems (seat belts) [9,10,11].

Recently, POM composites with antistatic properties have been widely used in electronic applications such as antistatic and electromagnetic interference (EMI) applications [9,12,13]. For instance, POM composites were utilized for fixing jigs for electronic devices, such as mobile phones and displays, where low electrical conductivity is required to achieve the antistatic properties. Without the antistatic properties of the jigs, the circuits, substrates, and semiconductors can be damaged owing to dust and impurities during fabrication and storage. Electrically conductive POM composites can be fabricated through the incorporation of conductive fillers such as carbon black, carbon fiber, graphite, and metals [14,15,16,17].

Carbon nanotubes (CNTs) have been extensively utilized owing to their extraordinary electrical and mechanical properties and large aspect ratio [18,19,20,21]. Thus, CNTs can replace conventional fillers in the fabrication of multifunctional polymeric nanocomposites [22,23]. In particular, CNT-embedded nanocomposites are ideal candidates for fabricating electrically conductive polymer nanocomposites [22,23]. However, a major drawback for utilizing CNTs is their low dispersion in polymer matrices because of their entanglement and aggregation into bundles [24,25]. This is chiefly caused by van der Waals and electrostatic forces of CNTs [26]. Although they can be slightly dispersed using only limited kinds of organic solvents, such as *N*-methylpyrrolidone (NMP), *o*-dichlorobenzene (ODCB), *N*,*N*-dimethylformamide (DMF), and *N*,*N*-dimethylacetamide (DMAc), the degrees of stability and isolation are insufficient for most applications. Moreover, POM is highly crystalline, which makes it more difficult to achieve a uniform dispersion of nanofillers in its matrix. Chemical functionalization, noncovalent wrapping, and blended matrix systems have been utilized to solve the dispersion problems [27,28,29].

In this study, we designed noncovalent wrapping systems using cyanuric acid and ethylene-co-acid-co-sodium acid copolymer-based ionomers to enhance the CNT dispersion in the POM matrix with a view to improving the electrical properties (electrical conductivity, surface resistance, and their monodispersion) of the nanocomposites.

## 2. Experimental

### 2.1. Materials

POM was supplied by BASF Co., (Augsburg, Germany, model: N2320, specific gravity: 1.410, melt flow index: 5.3 g/10 min), and multiwalled carbon nanotubes (MWCNTs) were purchased from Kumho Petrochemical Co., (Seoul, Korea, model: 210T, diameter: 11–13 nm, length: 40–50 µm, purity: 95 wt%, bulk density: 0.025 g/ML). Hydroxyl moieties were primarily detected on CNT surfaces (Appendix A). Ethylene-co-acid-co-sodium acid copolymer-based ionomer was supplied by Dow Chem Inc. (Midland, MI, USA, model: SURLYN 9910, specific gravity: 0.97, melt flow index: 0.7 g/10 min at 190 °C/2.16 kg, melting point: 86 °C). Cyanuric acid was purchased from Sigma–Aldrich (St. Louis, MO, USA).

### 2.2. Extrusion and Injection Processing

POM is vulnerable to acid hydrolysis and oxidation by mineral acid and chlorine [30,31]. It is also susceptible to alkaline attack and degradation in hot water, and extreme shearing effects at elevated temperature [32,33]. Thus, the extrusion conditions for POM/CNT nanocomposite fabrication are of great importance. POM was dried in an oven (ThermoStable OF-50, Daihan Scientific Co., Wonju, Korea) at 60 °C for 5 h prior to extrusion. POM, CNT, and additives (when added) were mixed in a plastic bag using a tumbler mixer. (BNOChem Co., Cheongju, Korea) Subsequently, the mixture was fed into the main hopper of an intermeshing corotation twin-screw extruder (STS25–44V–SF, Hankook EM Ltd., Gyeonggi, Korea) at a hopper temperature of 100 °C at a rate of approximately 4 kg/h. The screw diameter and length/diameter (L/D) were 25 mm and 44 mm, respectively. The two die hole sizes were 4 mm, and the die was kept at 165 °C. The screw was rotated at 240 rpm. The barrel temperatures were in the range of 100–170 °C. The measured temperature of the melted composite resin was 185–190 °C, which was slightly higher than the barrel temperatures owing to internal friction caused by shearing effects. The residual pressure in the barrel was maintained at 0.09 MPa. The melted resins were cooled in a water bath at room temperature (22–24 °C) and then pelletized using a strand pelletizer (HNP1, Hankook EM Ltd., Korea) equipped with a rotary knife (a diameter of 119 mm and width of 50 mm at 450 rpm) to produce cylinder-type composite pellets with approximately a diameter of 1 mm and length of 3 mm. The pelletized samples were then dried in an oven at 60 °C for 24 h. The composite pellets were inserted into the injection molding machine (LGH50N, LS Mtron Co., Anyang, Korea) to manufacture specimens for various analysis methods such as tensile and rheological properties and Izod impact strength. The screw diameter for the injection molding machine was 25 mm. The maximum injection pressure and maximum injected amount per cycle were 247 MPa and 45 g, respectively. The barrel temperatures were set to be 190–200 °C. The injection pressures were 6.9 and 4.9 MPa for tensile and Izod impact strength test specimens, respectively. The injected specimens were cooled in a mold at room temperature.

### 2.3. Characterization

#### 2.3.1. Scanning Electron Microscopy (SEM)

The morphologies of POM/CNT composites with and without additives were observed using scanning electron microscopy (SEM; Apro, FEI Co., Hillsboro, OR, USA) at an electron beam voltage of 10.0 kV and magnification of 10,000× and 25,000×. The fractured specimens for SEM examination were obtained from the Izod impact strength tests. The fractured surface was coated with a 5–10 nm thick gold layer using a sputter coater (Cressington 108 Auto Sputter Coater, Ted Pella Inc., Redding, CA, USA) prior to the SEM measurements.

#### 2.3.2. Differential scanning calorimetry (DSC)

Differential scanning calorimetry (DSC; DSC25, TA Instruments, New Castle, DE, USA) was carried out to determine the melting (*T_m_*) and crystallization (*T_c_*) temperatures of polymers and nanocomposites. Approximately 3–5 mg of each sample was placed in a hermetic aluminum pan and heated at a scanning rate of 10 °C/min under nitrogen purging (50 mL/min). The crystallinity (*χ_c_*) of POM and POM/CNT composites was calculated based on a 100% crystalline POM melting enthalpy of 251.8 J/g [34]. The second heating cycle was used for determining the transition temperatures by eliminating the thermal history.

#### 2.3.3. Thermogravimetric Analysis (TGA)

Thermogravimetric analysis (TGA; Perkin Elmer Co., Waltham, MA, USA) was performed, according to ISO 11358. Samples with a mass of 1.0–2.0 mg were heated from 50 °C to 500 °C at a heating rate of 10 °C/min, and then held at 500 °C for 10 min prior to cooling. The nitrogen purge had a flow rate of 20.0 mL/min at gas pressure of 2.2 bar. The degradation point (*T_d_*) of nanocomposites was determined, on the basis of 1% weight loss.

#### 2.3.4. Dynamic Mechanical Analysis (DMA)

Dynamic mechanical analysis (DMA; Discovery DMA 850, TA Instruments, New Castle, DE, USA) was conducted in the tensile mode. Rectangular specimens (approximately 35 mm long, 10 mm wide, and 4 mm thick) were prepared to measure the storage and loss moduli and loss tangent (tan δ). The glass transition temperature (*T*_g_) values of POM and its nanocomposites were determined based on the peaks of tan δ. The measurements were performed at a single frequency of 1 Hz and constant amplitude of 20 µm. The heating rate was set at 3 °C/min.

#### 2.3.5. Tensile Test

Uniaxial tensile deformation was performed using a universal testing machine (UTM; TD-012, Testone Co., Siheung, Korea) according to ISO 527-2 1A. The specimen cross-section had dimensions of 10 mm × 4 mm and the gauge length was 80 mm. The cell load capacity was 10 kN. Specimens were elongated at a constant cross head speed of 50 mm/min at room temperature (22–24 °C). The mean values of each sample were determined using five specimens.

#### 2.3.6. Izod Impact Strength Test

Notched Izod impact strength tests (WL2200D, Withlab Co., Anyang, Korea) were performed according to ISO 180 with rectangular dimensions of 4.0 × 10 × 80 mm. The notch depth, radius, and angle of the specimens were 2 mm, 0.25 ± 0.5 mm, and 45°, respectively. The radius of the hammer knife edge was 0.8 mm, and the hammer lift angle was 150°. The hammer velocity at the moment of impact was 3.46 m/s. The capacity of the tester was 60 kgf/cm. The mean values of each POM and nanocomposite were determined using seven specimens.

#### 2.3.7. Melt Flow Index (MFI) Test

Melt flow index (MFI) was measured by an MFI machine (WL1400SA, Withlab Co., Anyang, Korea) according to KS M3070 and ISO E1133. The diameter and length of the standard die orifice (nozzle) were 2.095 and 8 mm, respectively. The piston diameter was 9.5 mm. Samples were dried at 60 °C for 4 h to remove moisture absorbed in POM and nanocomposites prior to the measurements. The measurement temperature was 190 °C, with a load of 2.16 kg. Pellets were inserted into the piston and preheated for 5 min.

#### 2.3.8. Rheological Properties

Rheological measurements were carried out using a Discovery Hybrid HR-10 rheometer (TA Instruments., New Castle, DE, USA) in a parallel plate (φ = 25 mm, gap = 1 mm) configuration. A dynamic strain sweep was initially used to determine the linear viscoelastic regions. The strain was set at 1%, and the angular frequency was scanned from 10^0^ to 10^3^ s^−1^. The test temperature was 180 °C.

#### 2.3.9. Transmission Electron Microscopy (TEM)

Transmission electron microscopy (TEM, JEM-1400, JEOL, Tokyo, Japan) was performed at 120 kV to investigate the morphology and dispersion of CNTs in the POM matrix. Ultrathin sections were obtained using microtome (RMC, Leica, Wetzlar, Germany) with a diamond knife.

#### 2.3.10. Fourier Transform Infrared (FTIR) Spectroscopy

Fourier transform infrared (FTIR, Nicolet 6700, Thermo Fisher Scientific Co., Waltham, MA, USA) spectroscopy was performed in the attenuated total reflection (ATR) mode to detect chemical bonds. Each FTIR spectrum was recorded in a wavenumber region of 4000–600 cm^−1^ by carrying out 16 scans.

## 3. Results and Discussion

Electronic devices are susceptible to electrostatic discharge (ESD) produced by triboelectrification (friction) during handling, packaging, and transportation processes, potentially leading to significant damage [35,36]. To avoid the ESD, antistatic packaging is required with a low electrical resistance that prevents charge accumulation, thereby facilitating the flow of electric charges through the polymeric composites [37]. Electrically insulating materials generally exhibit a relatively high surface resistance of >10^11^ Ω/sq that impedes the electron flow through their surfaces. In contrast, materials with antistatic and dissipative properties have moderate electrical resistance of 10^4^–10^11^ Ω/sq, allowing for the electrical conduction [38].

Pristine POM is an electrically insulating polymer with a surface resistance of >10^11^ Ω/sq. MWCNTs were utilized to decrease the surface resistance of the POM/CNT nanocomposites. As shown in Figure 1, the incorporation of 0.5 wt% CNT into the pristine POM substantially decreased the surface resistance of the POM/CNT composite to 10^8^–10^9^ Ω/sq. Further reduction was observed until a plateau was reached beyond 1.5 wt% CNT when the surface resistance was saturated. Appendix A shows the electrical conductivities of POM/CNT nanocomposites, determined by their surface resistances. The electrical conductivities increased as a function of CNT loading until 1.5 wt% CNT and then showed a saturation point above 1.5 wt%. Appendix A shows SEM images of pristine POM and POM/CNT nanocomposites. Some aggregates were observed in the nanocomposites without additives.

The thermal properties of polymer nanocomposites are routinely investigated when nanofillers are incorporated into the polymer matrix. In this study, DSC and TGA were employed to examine the crystallization, melting behavior, and thermal stabilities of the nanocomposites. Figure 2a,b show the heating and cooling DSC traces, respectively. Table 1 summarizes the melting (*T_m_*), crystallization (*T_c_*) and degradation (*T_d_*) temperatures, melting (Δ*H_m_*) and crystallization (Δ*H_c_*) enthalpy, and crystallinity (*χ_c_*) of the neat POM and POM/CNT nanocomposites. The *T_m_* values increased as a function of CNT concentration owing to secondary interactions between the polymer matrix and nanofillers. The incorporation of CNTs also increased *T_c_* values because the CNT particles acted as nucleating agents up on cooling. However, the exothermic peak became broader as a function of CNT concentration, indicating that the crystallization time increased with increasing CNT loading. The crystallinity of POM slightly increased with increasing CNT content. Thus, the incorporation of CNT into the POM matrix influenced the crystallization temperatures, onset point, time, and crystallinity. TGA thermograms (Figure 2c) revealed that the thermal stability of the nanocomposites was maintained until a CNT loading of 1.5 wt%, with further addition of CNT reducing the thermal stability.

In addition to the thermal properties, the mechanical properties of nanofiller-embedded polymer composites are crucial for a wide range of applications [39]. Although the electrical properties of the nanocomposites were considerably enhanced, the fabricated nanofiller-embedded composites would not see widespread applications if their mechanical properties were poor. As evident from Figure 3a,b, the tensile strength and modulus of the POM/CNT composites increased, while the elongation at break gradually decreased (Figure 3c) as the CNT loading increased. However, the incorporation of nanofillers typically resulted in a reduction in the elongation at break and toughness. In contrast, the Izod impact strength of the composites was enhanced by the infiltration of CNTs into the POM matrix (Figure 4).

DMA is a powerful technique for investigating the viscoelastic behaviors of polymeric nanocomposites by observing mechanical responses as a function of temperature or frequency. The strain in the composite was measured through the application of sinusoidal stress, thereby determining the complex modulus. Figure 5 shows the DMA data of the POM/CNT nanocomposites. The storage modulus (*E’*) of the nanocomposite decreased with the incorporation of 0.5 wt% CNT, likely due to the chain scission caused by shear effects of the nanofillers during extrusion and injection molding processes [40]. However, the CNT-POM interactions compete with the chain scission, and the CNT-POM interactions became dominant for CNT loadings of >1.0 wt%, enhancing the storage modulus. The transition temperatures (*T**_α_*, *T_β_*, *T**γ*, and *T_g_*) of polymeric materials can be measured using various methods such as DMA, thermal mechanical analysis (TMA), DSC, and dielectric analysis (DEA). Among these techniques, DMA is the most precise method to determine the transitions of the polymeric materials including composites. The transition temperatures can be derived from the onset temperature of the reduction in the storage modulus and the peak temperatures in the loss modulus (*E”*) and tan δ. As shown in Table 2, the *T**_α_* values increased with increasing CNT loading, which can be ascribed to more CNT-POM interactions than the chain scission. In addition, the increased *T**_α_* values indicate the restrictive chain mobility at the related temperature range, owing to CNT-polymer matrix interactions. *T_β_* was increased by the incorporation of CNT into the POM matrix. Especially, 0.5 wt% CNT addition showed the highest *T_β_* value, whereas the *T**_α_* value gradually increased as a function of CNT concentration.

The infiltration of nanofillers can significantly influence the melt flow and viscosity of nanocomposites during composite and product processing [41,42,43]. The MFI method and a rheometer were utilized to investigate the rheological properties of POM/CNT composites. The MFI did not change with a 0.5 wt% CNT loading but decreased considerably with loading of >0.5 wt%, plateauing at a CNT loading of 1.5 wt% as shown in Figure 6. The complex viscosity of the POM/CNT composites increased and saturated at a 1.5 wt% CNT loading (Figure 7a). The shear storage (*G*’) and shear loss (*G*”) moduli of the nanocomposites both increased with increasing CNT loading. Furthermore, the slope of *G’*, as a function of angular frequency, decreased as the CNT content increased, alluding to the formation of a 3D network of the nanofillers under dynamic conditions.

Figure 7d shows a plot of *G*’ vs. *G*” at a frequency associated with Cole–Cole plots that are used in dielectric spectroscopy [44]. Modified Cole–Cole plots have previously been utilized to infer information regarding the microstructures of nanocomposites [45,46]. For example, *G’* becoming dominant over *G”* is related to an increased level of long-chain branching. The dashed guideline in Figure 7d represents *G’* = *G”*, and it is evident that *G’* became larger than *G”* with increasing nanofiller content, which represents that the rheological behavior of the POM/CNT nanocomposite changed from liquid-like (viscous characteristic) to solid-like (elastic characteristic) state. The slope decreased with increasing CNT content because of the change in the microstructure of the POM/CNT composites.

The POM/CNT nanocomposite with 1.5 wt% CNT loading (POM/C1.5) was chosen as the reference material because of its enhanced electrical, thermal, mechanical, and rheological properties. The ethylene-co-acid-co-sodium acid copolymer-based ionomer and cyanuric acid were utilized to adjust the surface resistance and its monodispersity of the POM/C1.5 nanocomposite as shown in Figure 8. The incorporation of the ionomer reduced the surface resistances of the composites, reaching a minimum at 3.0 wt% loading (POM/C1.5/I3) (Figure 8a). Cyanuric acid loading only slightly influenced the mean surface resistance of the nanocomposites (Figure 8b), with a loading of 0.5 wt% (POM/C1.5/A0.5) having the lowest surface resistance polydispersity. The surface resistance polydispersity as well as the mean surface resistance are crucial for antistatic and dissipative properties, especially for electronic device-fixing jigs. The combination of ionomer (3 wt%) and cyanuric acid (0.5 wt%) led to the lowest surface resistance with monodispersed-surface resistance. Appendix A shows the electrical conductivities of various nanocomposites, determined by their surface resistances. The proposed mechanisms for the microstructures of composites are illustrated in Figure 9. Covalent and noncovalent modifications can be used for surface treatments of CNTs [47,48]. The surface oxidation-induced carboxylic moieties and chemical reactions between CNTs and functional groups are common examples of covalent modifications. These covalent modifications give high stabilities of functionalization compared to the non-covalent modifications. The effective reinforcement of polymer composites can be achieved by a load transfer mechanism from the polymer matrix to the CNTs through the chemical bonding [49,50]. However, the conductivities and mechanical properties of CNTs are changed by the covalent modification and CNTs are often severed into short tubes [51]. In contrast, non-covalent modification provides the CNT surface treatment without deterioration in their inherent properties by facilely mixing additives and CNTs together under shear force or sonication [52]. Non-covalent functionalization can be achieved by two classifications of interaction mechanisms: enthalpy- and entropy-driven interactions [52]. Enthalpy-driven interactions include π-π, CH-π, NH-π, and cation-π between the additives and CNT surfaces, whereas entropy-driven interactions are involved with hydrophobic interactions using surfactants such as sodium dodecyl sulfate (SDS), sodium dodecylbenzene sulfonate (SDBS), sodium cholate (SC), and cetyltrimethylammonium bromide (CTAB) [53,54,55,56]. One of the examples for non-covalent functionalization, mainly related to enthalpy-driven functionalization, is the polymer wrapping of CNTs via various interactions. The polymer wrapping method was performed using various polymer types such as π-conjugated polymers, aromatic polymers, nonaromatic polymers (i.e., acrylate-based polymers), cationic polymers, block copolymers, and pendant polymers [57,58,59,60]. In this study, we employed each additive, including cyanuric acid (π-π interaction) and ionomer (cation-π interaction), and their combination to achieve the polymer wrapping on CNTs as shown in Figure 9.

Thermal properties of pristine POM and various nanocomposites are shown in Figure 10 and Appendix A and Table 3 and Table 4. The incorporation of ionomer only slightly influenced the *T_m_* values but increased the *T_c_* values. The crystallinity (*χ_c_*) of the composites slightly decreased with increasing loading of additives (each ionomer and cyanuric acid), because the additives plasticized the POM composites, thereby hindering the crystallization as shown in Table 3 and Appendix A. The POM/C1.5/A0.5 nanocomposite increased the *T_m_* value from 167.7 to 169.3 °C and decreased the *T_c_* value from 141.1 to 136.6 °C. The infiltration of ionomer and cyanuric acid into the POM/C1.5 nanocomposites enhanced the thermal stabilities of the nanocomposites owing to the interfacial interactions among the POM, CNT, and additives. However, excess cyanuric acid loading decreased the *T_d_* value of nanocomposites. The combination of ionomer and cyanuric acid for the POM/CNT nanocomposites balanced the thermal properties of the nanocomposites. SEM images of various nanocomposites as a function of each additive loading are shown in Appendix A.

The mechanical properties of CNT-infiltrated nanocomposites were mainly related to the CNT type, concentration, dispersity, and surface modification. Similar to POM/CNT nanocomposites, the. mechanical properties of pristine POM and various nanocomposites were examined using tensile and Izod impact strength tests as shown in Figure 11 and Appendix A. The tensile strength of the nanocomposites was slightly reduced, owing to the presence of each additive, whereas the tensile modulus of additive-embedded nanocomposites slightly improved. The elongation at break barely changed, whereas the Izod impact strength of nanocomposites decreased by the incorporation of additives. The Izod impact strength of nanocomposites gradually decreased as a function of each additive content as shown in Appendix A. The incorporation of additives typically reduced the impact strength of the nanocomposites. Thus, thermoplastic polyurethane (TPU) was employed to compensate for the loss in impact strength in this study. The infiltration of TPU into the nanocomposites enhanced the Izod impact strength from 4.2 to 5.6 kJ/m^2^.

The variations in the thermal transition of the POM nanocomposites upon the incorporation of additives were investigated using DMA. The thermal transitions were determined based on peaks in the *E”* and tan δ. The transition (*T_γ_*) of POM at approximately –60 °C is typically considered as the glass transition temperature (*T_g_*), and it is associated with the motion of short segments in the disordered regions of the POM chains. The broad peak at approximately 120 °C (*T**_α_* in the range of 50–150 °C) is characteristic of highly crystalline POM [61]. The transition is involved with translational motions of the crystalline structure along the chains. The incorporation of CNTs and combination of CNTs and additives into the nanocomposites barely affected the T_α_ values of the nanocomposites. The transition (*T_β_*) with a low intensity between –30 and 30 °C was ascribed to the motion of long segments in the disordered regions of the POM chains, low crystallinity, absorbed water, or thermal history of the composite. The *T_β_* value of the nanocomposites increased with increasing each additive (i.e., ionomer and cyanuric acid). The increased transition temperatures represent the restrictive chain mobility at high temperatures, owing to additive-induced CNT-polymer matrix interactions. The storage (*E’*) and loss (*E”*) moduli and the tan δ of the nanocomposites are shown in Figure 12, Appendix A and Table 5 and Table 6. The *E’* value of the nanocomposite decreased as a function of each additive. The reduction in *E’* was ascribed to the increment of the polymer chain mobility through the polymer-CNT interaction [62].

Understanding the rheology of a polymer composite is crucial for fabrication and processing including its extrusion and injection molding. The MFI measurement offers information regarding flowability (viscosity) in the medium shear rate region. The incorporation of CNTs substantially reduced the MFI, representing the increased viscosity, as shown in Figure 13. The ionomer-embedded POM/CNT nanocomposites increased as a function of ionomer concentration owing to slippage between the ionomer-coated CNTs and melted POM polymer chains. The MFI increased for a 0.5 wt% cyanuric acid loading compared to the POM/C1.5 nanocomposite without cyanuric acid because of the high dispersity of cyanuric acid-attached CNT through the POM matrix, which was demonstrated by the electrical properties. By contrast, the MFI of the cyanuric acid-embedded POM/CNT nanocomposites over 1.0 wt% decreased with increasing cyanuric acid loading, probably due to the strong physical interactions between melted POM chains and cyanuric acid-attached CNTs caused by the π–π interaction between CNT and cyanuric acid [63]. This indicates that the interaction factor was dominant over dispersion factors above a 1.0 wt% cyanuric acid loading. The combination of ionomer and cyanuric acid somewhat increased the MFI value. Storage and loss moduli and the complex viscosity of the nanocomposites were also examined using a rheometer as shown in Figure 14, Appendix A. The complex viscosity, *G’* and *G”*, of POM/C1.5/I nanocomposites decreased as a function of ionomer content (Appendix A), which is analogous to the trend observed for the MFI results. Appendix A shows that *G’* became smaller than *G”* with increasing ionomer content. This indicates that the rheological behavior of the nanocomposite changed from a solid-like to liquid-like state. The complex viscosity and *G’* decreased upon the incorporation of 0.5 wt% cyanuric acid but increased with greater cyanuric acid loadings as shown in Appendix A. These findings were also similar to the MFI results. *G’* became larger than *G”* with increasing cyanuric acid concentration, which indicates the increased elastic behavior. The complex viscosity, *G’*, and *G”* of the nanocomposites all decreased by the combined incorporation of both ionomer and cyanuric acid into POMC1.5 as shown in Figure 14.

Appendix A shows SEM micrographs of POM and various POM/CNT nanocomposites. The CNTs of the POM/C1.5 nanocomposite without additives aggregated (Appendix A), whereas the incorporation of additives improved the CNT dispersity in the POM matrix. In addition to SEM, visual observation of CNTs in polymer matrices was routinely investigated by TEM. Figure 15 shows the morphologies of POM/C1.5 and POM/C1.5/I3/A0.5 nanocomposites. Compared to the POM/C1.5 nanocomposite, a relatively homogeneous CNT dispersion in the POM matrix was achieved for the POM/C1.5/I3/A0.5 nanocomposite. This indicates that the combination of ionomer and cyanuric acid enhanced the MWCNT dispersity in the POM matrix, thereby increasing the electrical properties, such as the electrical conductivity and its monodispersity, despite a slight reduction in the mechanical properties caused by plasticization effects. The interfacial interactions between the CNTs and the POM matrix decreased, and the surface-to-surface interparticle distance decreased with increasing number of clusters (agglomerates), thereby reducing the dispersity of CNTs in the POM matrix. The FTIR spectra of POM/C1.5/I3/A0.5 nanocomposite and each component are given in Appendix A. The peaks indicate that the nanocomposite indeed comprised POM, CNT, ionomer, and cyanuric acid after extrusion processing.

## 4. Conclusions

To improve the electrical conductivity of POM, we investigated the incorporation of MWCNTs. However, the POM/CNT composites suffered from low dispersion in the polymer matrix, thereby leading to high electrical conductivity and its poor monodispersity. To address this issue, we incorporated the additives of ionomer and/or cyanuric acid into the POM/CNT nanocomposites using a twin-screw extruder. The mechanical, thermal, and rheological properties of the POM/CNT nanocomposites were examined, in particular their surface resistance for the potential application as an electronic device-fixing jig. The effect of the ionomer was to enhance the electrical conductivity, whereas that of the ethylene-co-acid-co-sodium acid copolymer-based ionomer was to stabilize the electrical conductivity value (electrical conductivity monodispersity). The influences of not only each additive but also additive combinations were investigated. The tensile elongation at break and Izod impact strength of the nanocomposites were slightly reduced by the infiltration of additives into the POM/CNT nanocomposites, whereas their strength and modulus barely changed. The tensile strength of ionomer-embedded nanocomposites gradually decreased as a function of ionomer content. The incorporation of ionomer (forming a coating on CNTs) and/or cyanuric acid (stabilizing π-π interaction between CNTs and cyanuric acid) additives into the POM/CNT nanocomposites enhanced the CNT dispersion in the POM matrix, thereby improving the electrical properties such as the electrical conductivity and electrical conductivity monodispersity. The POM/C1.5/I3/A0.5 nanocomposite was determined to have the optimum composition. Nanocomposites with tunable electrical properties can be used, especially for antistatic and EMI applications such as electronic device-fixing jigs.

## Figures and Tables

**Figure 1 polymers-14-01849-f001:**
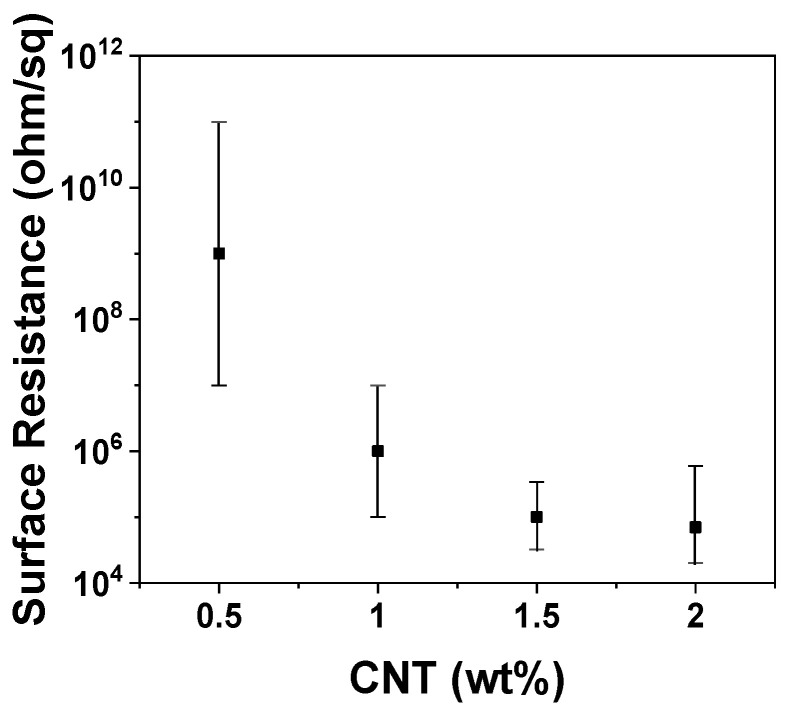
Surface resistance of POM/CNT composites as a function of CNT loading.

**Figure 2 polymers-14-01849-f002:**
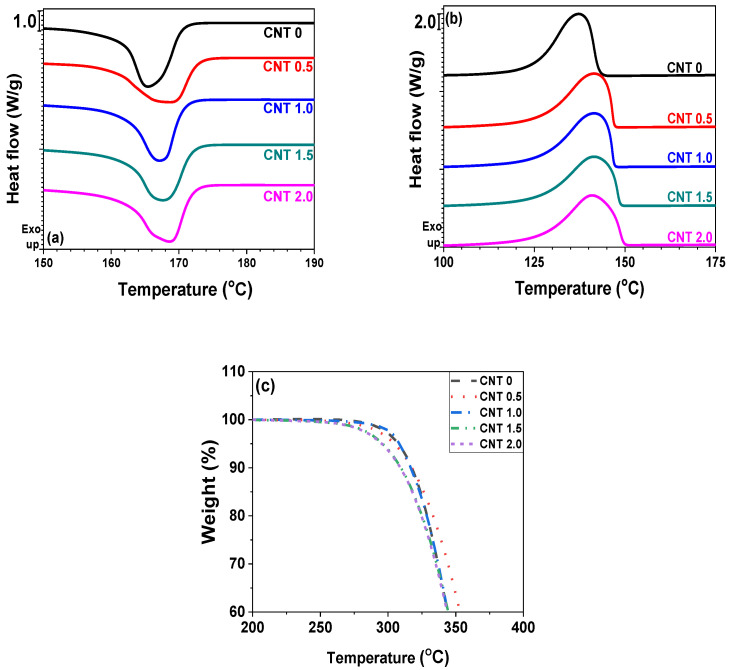
Thermal properties of pristine POM and POM/CNT composites with different CNT contents: (**a**) DSC heating; (**b**) DSC cooling traces; (**c**) TGA thermograms.

**Figure 3 polymers-14-01849-f003:**
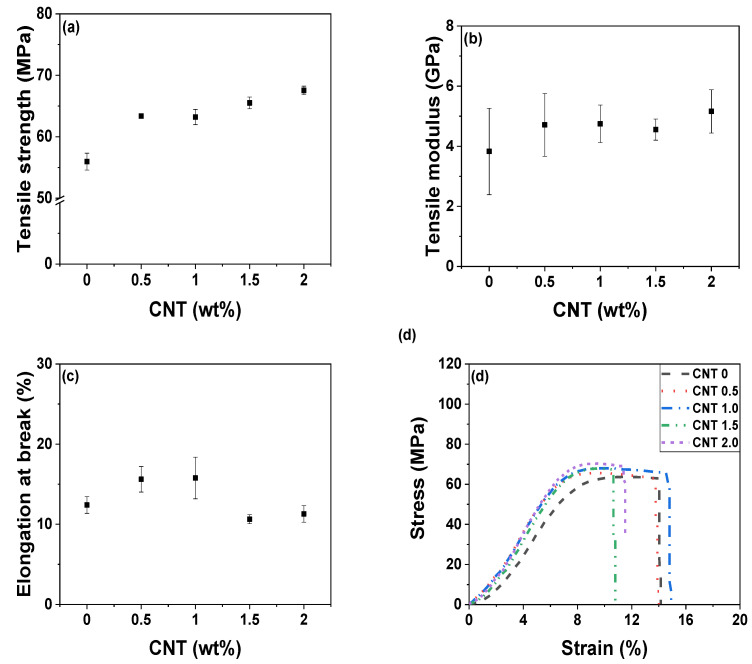
Tensile properties of POM/CNT nanocomposites: (**a**) ultimate tensile strength; (**b**) Young’s modulus; (**c**) elongation at break; (**d**) stress–strain curves.

**Figure 4 polymers-14-01849-f004:**
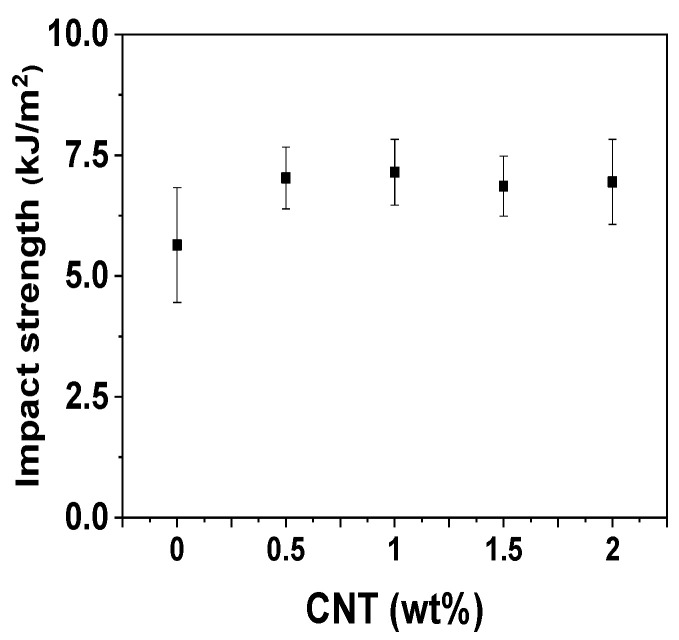
Izod impact strengths of POM/CNT nanocomposites.

**Figure 5 polymers-14-01849-f005:**
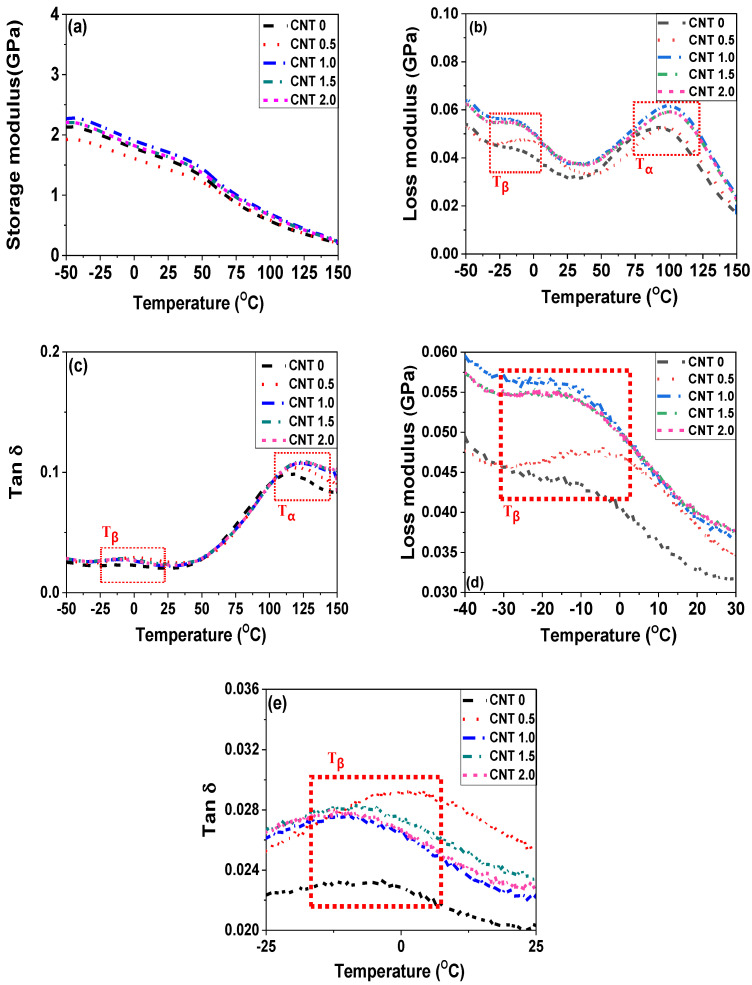
DMA data of POM/CNT nanocomposites: (**a**) storage modulus; (**b**,**d**) loss modulus; (**c**,**e**) tan δ; (**d**) zoomed in area of Figure 5b; (**e**) zoomed in area of Figure 5c.

**Figure 6 polymers-14-01849-f006:**
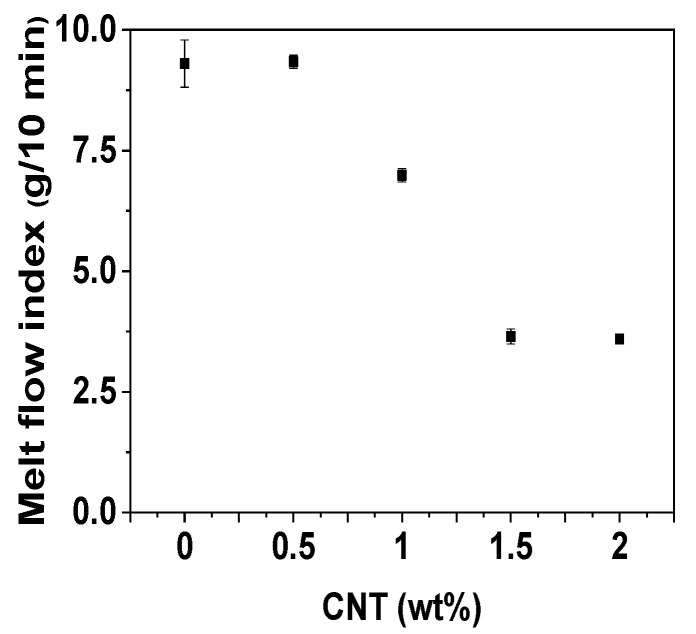
MFI of POM/CNT nanocomposites as a function of CNT concentration.

**Figure 7 polymers-14-01849-f007:**
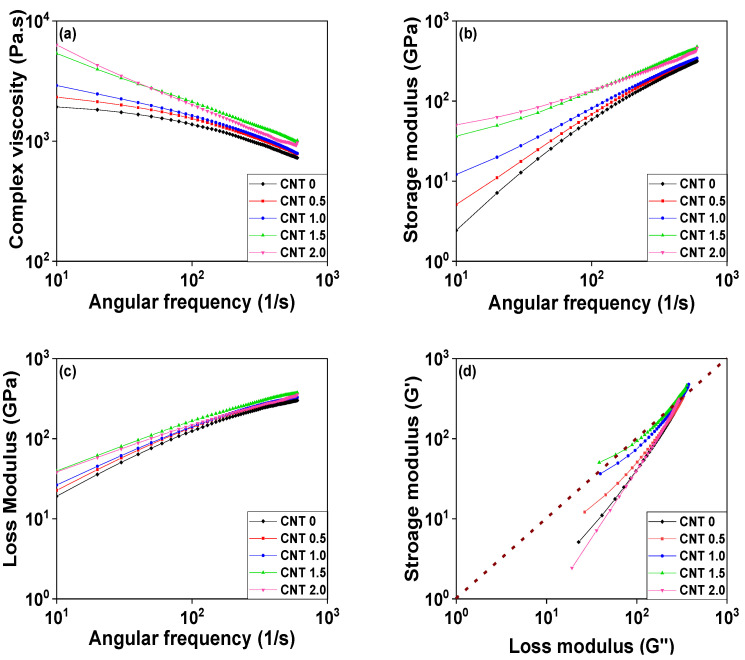
Rheological properties of the POM/CNT nanocomposites: (**a**) complex viscosity; (**b**) shear storage modulus (*G’*); (**c**) shear loss modulus (*G”*) as a function of frequency; (**d**) *G’* vs. *G”*.

**Figure 8 polymers-14-01849-f008:**
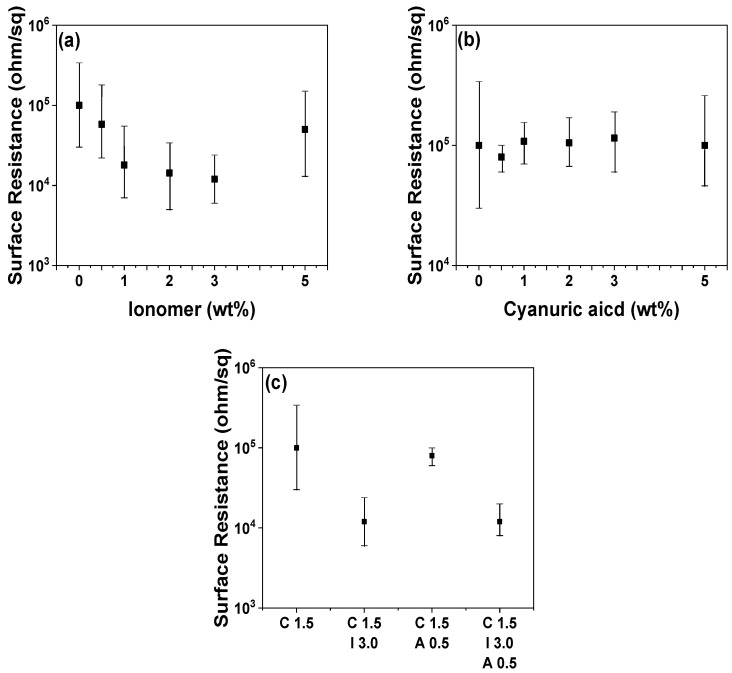
Surface resistances of neat POM/CNT (POM/C1.5), (**a**) POM/CNT/ionomer (POM/C1.5/I3), (**b**) POM/CNT/cyanuric acid (POM/C1.5/A0.5), and (**c**) POM/CNT/ionomer/cyanuric acid (POM/C1.5/I3/A0.5) composites with different additive concentrations.

**Figure 9 polymers-14-01849-f009:**
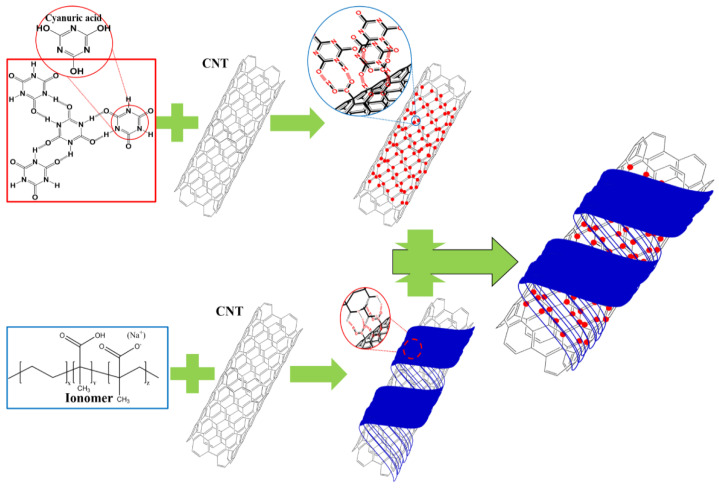
Schematic structures of POM/CNT/ionomer/cyanuric acid nanocomposite. The red and blue colors indicate cyanuric acid and ionomer, respectively.

**Figure 10 polymers-14-01849-f010:**
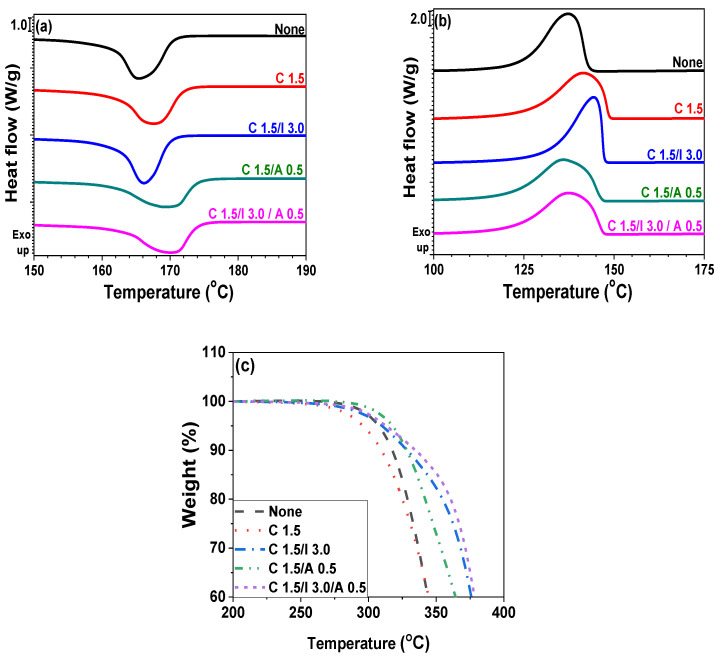
Thermal properties of pristine POM, POM/CNT (POM/C1.5), POM/CNT/ionomer (POM/C1.5/I3), POM/CNT/cyanuric acid (POM/C1.5/A0.5), and POM/CNT/ionomer/cyanuric acid (POM/C1.5/I3/A0.5) composites: (**a**) DSC heating and (**b**) cooling traces; (**c**) TGA thermograms.

**Figure 11 polymers-14-01849-f011:**
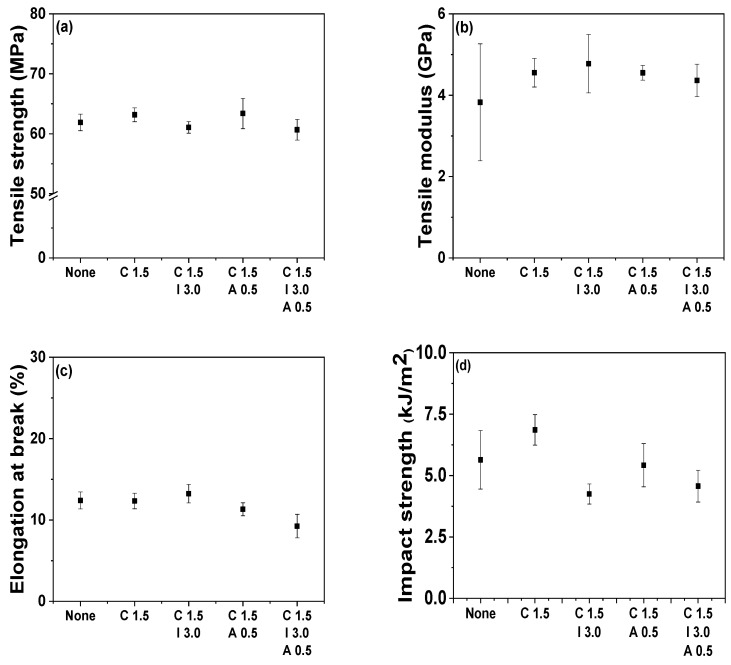
Tensile properties (**a**–**c**) and Izod impact strength (**d**) of POM/C1.5, POM/C1.5/I, POM/C1.5/A, and POM/C1.5/I3/A0.5 nanocomposites: (**a**) tensile strength; (**b**) tensile modulus; (**c**) elongation at break; (**d**) Izod impact strength.

**Figure 12 polymers-14-01849-f012:**
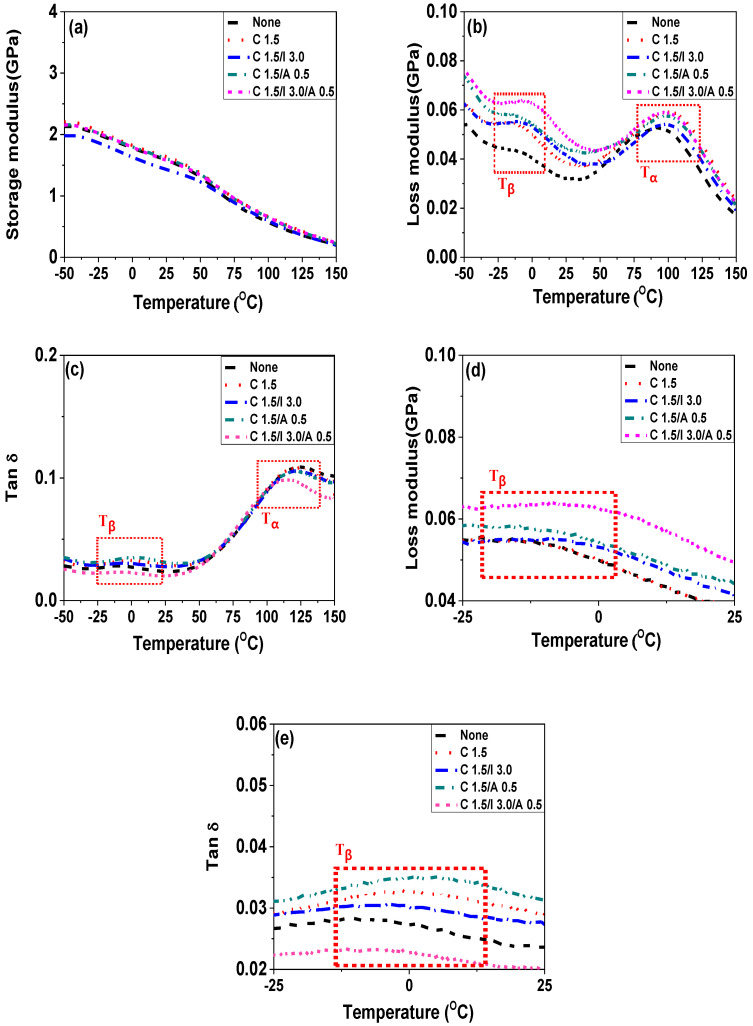
DMA data of neat POM and various nanocomposites (POM/C1.5, POM/C1.5/I3, POM/C1.5/A0.5, and POM/C1.5/I3/A0.5): (**a**) storage modulus; (**b**,**d**) loss modulus; (**c**,**e**) tan δ; (**d**) zoomed in area of Figure 12b; (**e**) zoomed in area of Figure 12c.

**Figure 13 polymers-14-01849-f013:**
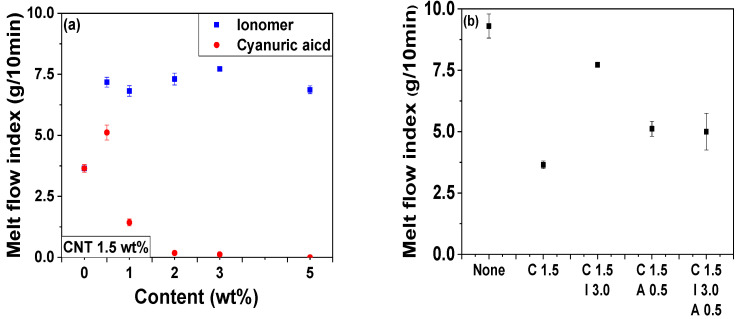
MFI of POM, POM/C1.5, POM/C1.5/I, POM/C1.5/A, and POM/C1.5/I/A nanocomposites with different additive concentrations: (**a**) POM/C1.5; POM/C1.5/I; POM/C1.5/A; (**b**) pristine POM and POM/C1.5/I/A.

**Figure 14 polymers-14-01849-f014:**
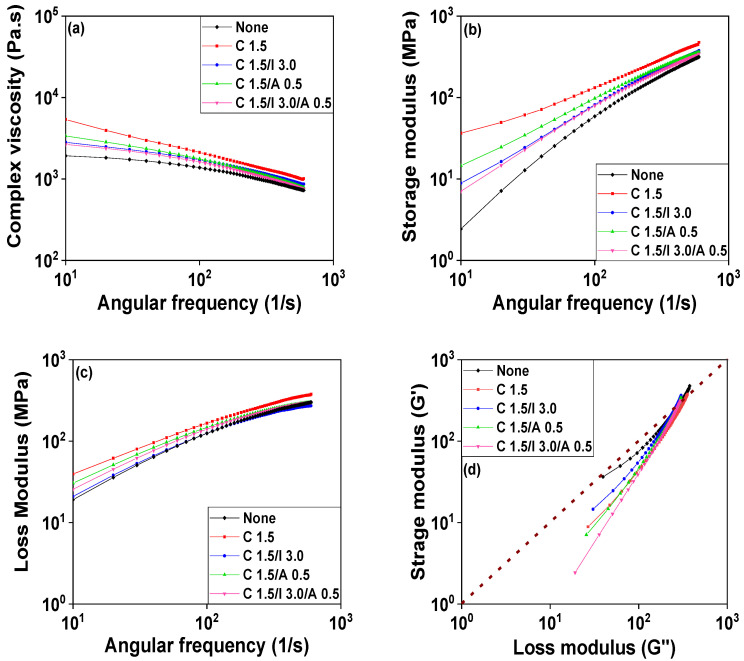
Rheological properties of POM, POM/C1.5, POM/C1.5/I3, POM/C1.5/A0.5, and POM/C1.5/I3/A0.5 nanocomposites: (**a**) complex viscosity; (**b**) shear storage modulus (*G’*); (**c**) shear loss modulus (*G”*) as a function of frequency; (**d**) *G’* vs. *G”*.

**Figure 15 polymers-14-01849-f015:**
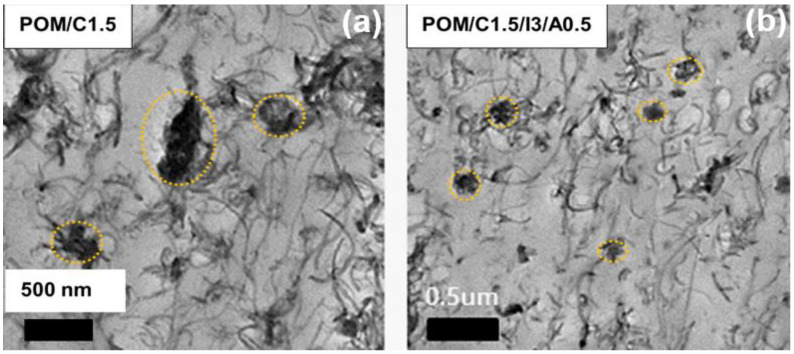
TEM images of (**a**) POM/C1.5, and (**b**) POM/C1.5/I3/A0.5 nanocomposites.

**Table 1 polymers-14-01849-t001:** Thermal properties of POM/CNT composites with different CNT contents.

CNT (wt%)	0	0.5	1.0	1.5	2.0
*T_m_* (°C)	165.4	167.0	165.7	167.8	168.6
Δ*H_m_* (J/g)	159.9	157.5	159.4	161.4	170.8
*T_c_* (°C)	137.2	141.0	142.8	141.1	140.9
Δ*H_c_* (J/g)	153.9	153.6	148.2	153.0	159.6
*χ_c_* (%) *	63.5	62.2	62.7	63.2	66.5
*T_d_* (°C)	287.3	287.2	287.8	267.8	266.1

* One hundred percent crystalline POM enthalpy: 251.8 J/g.

**Table 2 polymers-14-01849-t002:** Transition temperatures of neat POM and POM/CNT nanocomposites as a function of CNT concentration, based on DMA data.

CNT (wt%)		0.0	0.5	1.0	1.5	2.0
*T_β_* (°C)	*G*”	−30.6	−4.3	−30.1	−22.1	−22.5
	Tan δ	−11.7	1.5	−9.6	−8.5	−13.2
*T**_α_* (°C)	*G*”	93.3	97.8	100.5	99.9	98.3
	Tan δ	114.8	120.8	123.1	125.7	125.1

**Table 3 polymers-14-01849-t003:** Thermal properties of POM/C1.5, POM/C1.5/ionomer, and POM/C1.5/cyanuric acid composites as a function of additive concentration.

Content (wt%) *		0.0	0.5	1.0	2.0	3.0	5.0
*T_m_* (°C)	Ionomer	167.7	166.6	168.0	167.2	166.1	166.3
	Cyanuric acid		169.3	167.8	167.0	167.6	168.1
∆*H_m_* (J/g)	Ionomer	161.4	168.7	181.3	164.2	159.7	181.0
	Cyanuric acid		157.4	152.1	182.2	178.7	160.5
*T_c_* (°C)	Ionomer	141.1	142.2	142.83	140.2	148.0	143.1
	Cyanuric acid		136.6	139.3	137.2	139.7	133.6
∆*H_c_* (J/g)	Ionomer	153.0	155.9	175.5	152.6	143.5	163.4
	Cyanuric acid		149.8	138.4	169.6	166.0	151,1
*χ_c_* (%) *	Ionomer	63.2	65.6	70.2	62.9	60.6	67.2
	Cyanuric acid		61.3	58.9	69.8	67.8	59.6
*T_d_* (°C)	Ionomer	267.8	285.7	280.1	279.9	275.7	284.2
	Cyanuric acid		296.9	293.7	289.7	250.7	253.6

* One hundred percent crystalline POM enthalpy: 251.8 J/g.

**Table 4 polymers-14-01849-t004:** Thermal properties of pristine POM, POM/CNT (POM/C1.5), POM/CNT/ionomer (POM/C1.5/I3), POM/CNT/cyanuric acid (POM/C1.5/A0.5), and POM/CNT/ionomer/cyanuric acid (POM/C1.5/I3/A0.5) nanocomposites.

Content (wt%)	None	C1.5	C1.5 I3.0	C1.5 A0.5	C1.5 I3.0 A0.5
*T_m_* (°C)	165.4	167.7	166.2	169.3	170.2
∆*H_m_* (J/g)	159.9	161.4	159.7	157.4	149.8
*T_c_* (℃)	137.2	141.1	148.0	136.6	150.3
∆*H_c_* (J/g)	153.9	153.0	143.5	149.8	137.4
*χ_c_* (%) *	63.5	63.2	60.6	61.3	56.5
*T_d_* (°C)	287.3	267.8	275.7	296.9	281.8

* One hundred percent crystalline POM enthalpy: 251.8 J/g.

**Table 5 polymers-14-01849-t005:** Transition temperatures of POM/C1.5, POM/C1.5/I, and POM/C1.5/A nanocomposites with different additive concentrations, based on DMA data.

Contents (wt%)			0.0	0.5	1.0	2.0	3.0	5.0
*T_β_* (°C)	*G”*	Ionomer	−22.1	−21.3	−33.5	−10.9	−14.3	−28.5
		Cyanuric acid	−22.1	−24.9	−12.8	−5.6	−0.43	0.6
	Tan δ	Ionomer	−8.8	−8.4	−12.5	−0.1	1.2	−5.2
		Cyanuric acid	−8.8	−4.1	1.3	3.7	4.7	9.5
*T**_α_* (°C)	*G”*	Ionomer	99.9	100.0	100.7	97.9	97.3	95.5
		Cyanuric acid	99.9	99.2	99.7	97.3	97.8	96.4
	Tan δ	Ionomer	125.7	120.3	123.1	123.6	121.2	122.3
		Cyanuric acid	125.7	120.2	123.0	119.3	119.4	118.9

**Table 6 polymers-14-01849-t006:** Transition temperatures of POM, POM/C1.5, POM/C1.5/I3, POM/C1.5/A0.5, and POM/C1.5/I3/A0.5 nanocomposites with different additive concentrations, based on DMA data.

Contents (wt%)		None	C1.5	C1.5 I3.0	C1.5 A0.5	C1.5 I3.0 A0.5
*T_β_* (°C)	*G”*	−30.6	−22.1	−14.3	−24.9	−7.4
	Tan δ	−11.7	−8.5	1.2	−4.1	−11.7
*T**_α_* (°C)	*G”*	93.3	99.9	98.0	100.0	97.5
	Tan δ	114.8	125.7	123.1	125.7	122.0

## Data Availability

Not applicable.

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
