# Peer review of "Influence of Ionomer and Cyanuric Acid on Antistatic, Mechanical, Thermal, and Rheological Properties of Extruded Carbon Nanotube (CNT)/Polyoxymethylene (POM) Nanocomposites"

_polymers, 2022, doi:10.3390/polym14091849_

Round 1

Reviewer 1 Report

The manuscript is devoted to development of antistatic compositions based on POM, CNT and some active additives, their nature, the order of introducing, the dispersity degree and corresponding properties are the main results of this research. The main original additive is ionomer, which authors denote as (ethylene–co–acid–co–sodium acid) copolymer. The second one is traditional melamine enriched with amine groups. The idea was to couple (or to coat) ionomer and melamine molecules on CNT surface accepting that their presence can improve antistatic properties due to ionomer coating on CNT and melamine π-π interaction with CNT improving the CNT dispersibility in the POM matrix. The main results consists of enhancing the electrical conductivity and its monodispersity.  Testing of nanocomposites obtained by different physical methods, authors made choice in favor of optimal content of each component for antistatic and electromagnetic interference applications.

Some questions and notes to authors, starting with binary systems POM-CNT:

  • What functional groups are located in multiwall nanotubes surface? It is possible to suppose that mainly hydroxyl groups coupling with carboxy groups of ionomer or amine groups of melamine, but it should be proven.
  • Instead “vacuum pressure” better to use “the residual pressure” (page 3). The same concerns such units as kgf/cm2 and Ω/sq (next page).
  • Beginning the chapter “Results and discussion” with figure is not a good idea. Better to move it below.
  • Concerning data of mechanical tests, I am not sure that all these graphs should be shown here. I suggest to thin out a number of the same type of figures. Some results can be described in the text.
  • Discussion about concurrence of chain scission and POM-CNT interaction (pp 9-10) look like hypothesis.
  • Enumeration of different relaxation transitions registered by DMA method is in excess. Indeed, authors have used only two temperatures (table 2) without reasonable explanation of their nature).
  • Concerning rheological data (page 12), a tendency exists of transition to yielding behavior. It is reasonable to discuss these data, including Cole-Cole plot from these positions.
  • On page 14 the unexpected transition to testing solid materials including additives appears. It seems to me, that it should be precede with description of the final materials compositions and then figures. To be honest, I expected more strong influence of additives on surface resistance.
  • I do not like such distribution of results. Authors started with PON-CNT composite and then, using it as the reference sample, introduced additives, repeating the same order of methods and the same type of figures. I prefer to consider a role of additives in step-by-step manner. But approach used in the manuscript is also has right on life.
  • Unfortunately, the most intrigue picture shown in figure 9 was not discussed as whole. The blue tapes of CNT-ionomer complexes remained without explanation.
  • SEM micrographs (figure 15) did not show anything interesting. I would suggest to authors remain only TEM data, where improving of dispersity id possible to see.

Resume: authors carried out a great work with more-less positive results because the output characteristics of “novel” composites were improved a little bit only. Nevertheless, I think that this paper can be published in Polymers. Concerning my comments, many of them are advices, but the manuscript became better if authors will follow them.          

The manuscript is devoted to development of antistatic compositions based on POM, CNT and some active additives, their nature, the order of introducing, the dispersity degree and corresponding properties are the main results of this research. The main original additive is ionomer, which authors denote as (ethylene–co–acid–co–sodium acid) copolymer. The second one is traditional melamine enriched with amine groups. The idea was to couple (or to coat) ionomer and melamine molecules on CNT surface accepting that their presence can improve antistatic properties due to ionomer coating on CNT and melamine π-π interaction with CNT improving the CNT dispersibility in the POM matrix. The main results consists of enhancing the electrical conductivity and its monodispersity.  Testing of nanocomposites obtained by different physical methods, authors made choice in favor of optimal content of each component for antistatic and electromagnetic interference applications.

Some questions and notes to authors, starting with binary systems POM-CNT:

  • What functional groups are located in multiwall nanotubes surface? It is possible to suppose that mainly hydroxyl groups coupling with carboxy groups of ionomer or amine groups of melamine, but it should be proven.
  • Instead “vacuum pressure” better to use “the residual pressure” (page 3). The same concerns such units as kgf/cm2 and Ω/sq (next page).
  • Beginning the chapter “Results and discussion” with figure is not a good idea. Better to move it below.
  • Concerning data of mechanical tests, I am not sure that all these graphs should be shown here. I suggest to thin out a number of the same type of figures. Some results can be described in the text.
  • Discussion about concurrence of chain scission and POM-CNT interaction (pp 9-10) look like hypothesis.
  • Enumeration of different relaxation transitions registered by DMA method is in excess. Indeed, authors have used only two temperatures (table 2) without reasonable explanation of their nature).
  • Concerning rheological data (page 12), a tendency exists of transition to yielding behavior. It is reasonable to discuss these data, including Cole-Cole plot from these positions.
  • On page 14 the unexpected transition to testing solid materials including additives appears. It seems to me, that it should be precede with description of the final materials compositions and then figures. To be honest, I expected more strong influence of additives on surface resistance.
  • I do not like such distribution of results. Authors started with PON-CNT composite and then, using it as the reference sample, introduced additives, repeating the same order of methods and the same type of figures. I prefer to consider a role of additives in step-by-step manner. But approach used in the manuscript is also has right on life.
  • Unfortunately, the most intrigue picture shown in figure 9 was not discussed as whole. The blue tapes of CNT-ionomer complexes remained without explanation.
  • SEM micrographs (figure 15) did not show anything interesting. I would suggest to authors remain only TEM data, where improving of dispersity id possible to see.

Resume: authors carried out a great work with more-less positive results because the output characteristics of “novel” composites were improved a little bit only. Nevertheless, I think that this paper can be published in Polymers. Concerning my comments, many of them are advices, but the manuscript became better if authors will follow them.          

The manuscript is devoted to development of antistatic compositions based on POM, CNT and some active additives, their nature, the order of introducing, the dispersity degree and corresponding properties are the main results of this research. The main original additive is ionomer, which authors denote as (ethylene–co–acid–co–sodium acid) copolymer. The second one is traditional melamine enriched with amine groups. The idea was to couple (or to coat) ionomer and melamine molecules on CNT surface accepting that their presence can improve antistatic properties due to ionomer coating on CNT and melamine π-π interaction with CNT improving the CNT dispersibility in the POM matrix. The main results consists of enhancing the electrical conductivity and its monodispersity.  Testing of nanocomposites obtained by different physical methods, authors made choice in favor of optimal content of each component for antistatic and electromagnetic interference applications.

Some questions and notes to authors, starting with binary systems POM-CNT:

  • What functional groups are located in multiwall nanotubes surface? It is possible to suppose that mainly hydroxyl groups coupling with carboxy groups of ionomer or amine groups of melamine, but it should be proven.
  • Instead “vacuum pressure” better to use “the residual pressure” (page 3). The same concerns such units as kgf/cm2 and Ω/sq (next page).
  • Beginning the chapter “Results and discussion” with figure is not a good idea. Better to move it below.
  • Concerning data of mechanical tests, I am not sure that all these graphs should be shown here. I suggest to thin out a number of the same type of figures. Some results can be described in the text.
  • Discussion about concurrence of chain scission and POM-CNT interaction (pp 9-10) look like hypothesis.
  • Enumeration of different relaxation transitions registered by DMA method is in excess. Indeed, authors have used only two temperatures (table 2) without reasonable explanation of their nature).
  • Concerning rheological data (page 12), a tendency exists of transition to yielding behavior. It is reasonable to discuss these data, including Cole-Cole plot from these positions.
  • On page 14 the unexpected transition to testing solid materials including additives appears. It seems to me, that it should be precede with description of the final materials compositions and then figures. To be honest, I expected more strong influence of additives on surface resistance.
  • I do not like such distribution of results. Authors started with PON-CNT composite and then, using it as the reference sample, introduced additives, repeating the same order of methods and the same type of figures. I prefer to consider a role of additives in step-by-step manner. But approach used in the manuscript is also has right on life.
  • Unfortunately, the most intrigue picture shown in figure 9 was not discussed as whole. The blue tapes of CNT-ionomer complexes remained without explanation.
  • SEM micrographs (figure 15) did not show anything interesting. I would suggest to authors remain only TEM data, where improving of dispersity id possible to see.

Resume: authors carried out a great work with more-less positive results because the output characteristics of “novel” composites were improved a little bit only. Nevertheless, I think that this paper can be published in Polymers. Concerning my comments, many of them are advices, but the manuscript became better if authors will follow them.          

Author Response

Please find the attached file as a response.

Reviewer 2 Report

Before acceptance of this paper, the authors must address the following comments:

a) Please clarify the paragraph of lines 37 and 38.

b) Please provide information about the pelletized process, as well as pellets morphology.

c) Line:120: The authors must provide information about the cell load capacity of the universal testing machine. 

d) The authors need to provide the definition of some terms listed in Table 1 and the relevance of their values in the material composite behavior.

e) The authors need better arrangement of images in each manuscript Figure so that the reader can follow the information provided.

f)Figure 3: (a) Ultimate tensile strength, (b) Young Modulus? Please clarify.

g) Line 229. Please identify the transition temperatures Tα, Tβ, and Tγ in Figures 5.

h) Figure 5e is a zoom in view of Figure 5c? Please clarify.

i)Figure 7d: The authors must show the slope of each curve and explain its connection to the composite material crosslinked density.

j) Caption of Figure 8 must be improved.

k) Line 335: Please define TPU . Thermoplastic polyurethane?

L) Please revise line 390-393. This paragraph is not clear.

m) Please connect Figure 14 d) with composite crosslink density properties.

n) Figure 16. Please explain the cluster formation exhibited in TEM images a and b. The authors must provide insightful information of how this clusters influence the composite material behavior.

o) Figure S16 must be part of the main text. Please make sure to connect this figure with the crystallization properties of the composite materials.

p) The authors provide information about surface resistance of some of their developed composite materials however, I did not see any information regarding electrical conductivity. Please add this information in the modified manuscript version.

q) Line 453: Please provide a clear example of how the material must be tuned.

r) Please fix grammar and typo errors.

Author Response

Please find an attached file as a response.
